# Healthcare utilization during the first two waves of the COVID-19 epidemic in South Africa: A cross-sectional household survey

Nicole Wolter[1,2]*, Stefano Tempia[1,3], Anne von Gottberg[1,2], Jinal N. Bhiman[1,2], Sibongile Walaza[1,3], Jackie Kleynhans[1,3], Jocelyn Moyes[1,3], Sue Aitken[4], Sarah Magni[3,4], Jessica Yun[4], Tamika Fellows[4], Tetelo Makamadi[4], Renay Weiner[3], Cherie Cawood[5], Neil Martinson[6,7], Limakatso Lebina[6], Cheryl Cohen[1,3]

1 Centre for Respiratory Diseases and Meningitis, National Institute for Communicable Diseases (NICD) of the National Health Laboratory Service, Johannesburg, South Africa, 2 School of Pathology, Faculty of Health Sciences, University of the Witwatersrand, Johannesburg, South Africa, 3 School of Public Health, Faculty of Health Sciences, University of the Witwatersrand, Johannesburg, South Africa, 4 Genesis Analytics, Johannesburg, South Africa, 5 Epicentre Health Research, Durban, South Africa, 6 Perinatal HIV Research Unit (PHRU), University of the Witwatersrand, Johannesburg, South Africa, 7 Johns Hopkins University Center for TB Research, Baltimore, Maryland, United States of America

* nicolew@nicd.ac.za

**Data Availability Statement:** The dataset used and analysed during the current study is available at

## Abstract

Healthcare utilization surveys contextualize facility-based surveillance data for burden estimates. We describe healthcare utilization in the catchment areas for sentinel site healthcare facilities during the first year of the COVID-19 pandemic. We conducted a cross-sectional healthcare utilization survey in households in three communities from three provinces (KwaZulu-Natal, Western Cape and North West). Field workers administered structured questionnaires electronically with the household members reporting influenza-like illness (ILI) in the past 30 days or severe respiratory illness (SRI) since March 2020. Multivariable logistic regression was used to identify factors associated with healthcare utilization among individuals that reported illness. From November 2020 through April 2021, we enrolled 5804 households and 23,003 individuals. Any respiratory illness was reported by 1.6% of individuals; 0.7% reported ILI only, 0.8% reported SRI only, and 0.1% reported both ILI and SRI. Any form of medical care was sought by 40.8% (95% CI 32.9% - 49.6%) and 71.3% (95% CI 63.2% - 78.6%) of individuals with ILI and SRI, respectively. On multivariable analysis, respiratory illness was more likely to be medically attended for individuals at the Pietermaritzburg site (aOR 3.2, 95% CI 1.1–9.5, compared to Klerksdorp), that were underweight (aOR 11.5, 95% CI 1.5–90.2, compared to normal weight), with underlying illness (aOR 3.2, 95%CI 1.2–8.5), that experienced severe illness (aOR 4.8, 95% CI 1.6–14.3) and those with symptom duration of ≥10 days (aOR 7.9, 95% CI 2.1–30.2, compared to <5 days). Less than half of ILI episodes and only 71% of SRI episodes were medically attended during the first two COVID-19 waves in South Africa. Facility-based data may underestimate disease burden during the COVID-19 pandemic.

https://github.com/crdm-nicd/huts_hus_2020_2021.git".

**Funding:** We acknowledge funding from the South African Medical Research Council (https://www.samrc.ac.za/) (Reference number SHIPNCD 76756) [C. Cohen], The Wellcome Trust and the United Kingdom Foreign, Commonwealth and Development Office (https://wellcome.org/) (Grant no 221003/Z/20/Z) [C. Cohen] and United States Centers for Disease Control and Prevention (https://www.cdc.gov/) (Grant number 5 U01IP001048-05-00) [C. Cohen]. The funders had no role in study design, data collection and analysis, decision to publish, or preparation of the manuscript.

**Competing interests:** CC has received grant support from Sanofi Pasteur, South African Medical Research Council, The Wellcome Trust and the United Kingdom Foreign, Commonwealth and Development Office, PATH, US Centers for Disease Control and Prevention (CDC) and the Bill and Melinda Gates Foundation. NW and AvG have received grant support from Sanofi Pasteur, US Centers for Disease Control and Prevention (CDC) and the Bill and Melinda Gates Foundation. This does not alter our adherence to PLOS ONE policies on sharing data and materials.

## Introduction

The first case of severe acute respiratory syndrome coronavirus 2 (SARS-CoV-2) in South Africa was reported on 5 March 2020. A national lockdown was instituted from 27 March– 1 May 2020, which was followed by gradual and phased changing of restrictions in response to the national number of cases [1]. By the end of April 2021, South Africa had experienced two waves of infection with the first wave peaking in July 2020, and a second wave peaking in January 2021.

Healthcare utilization surveys (HUS) are useful to characterize the healthcare seeking patterns for diseases of interest. They have been used to complement sentinel surveillance by allowing assessment of the sensitivity of a surveillance system and adjustment of facility-based surveillance data to estimate disease burden within the community [2, 3]. Understanding healthcare utilization in a community helps to characterize healthcare seeking behaviour, strengthen access to healthcare and improve the interpretation of surveillance data. HUS were conducted in 2012 in Soweto and Klerksdorp communities in South Africa, approximately one-third of individuals with pneumonia did not seek care with a licensed service provider and all individuals with influenza-like illness (ILI) sought care [4]. HUS were conducted in Pietermaritzburg in 2013, 4% of individuals with pneumonia and 13% of individuals with ILI did not seek care [5]. A cross-sectional study in Johannesburg in 2015 showed that 32% of participants that reported ILI and 0% of participants that reported pneumonia did not seek care [6]. However, patterns in healthcare utilization changed during the coronavirus disease 2019 (COVID-19) pandemic, given lockdown restrictions and public fear for the then novel virus [7].

The National Institute for Communicable Diseases (NICD) has conducted syndromic surveillance for severe respiratory illness (SRI) and influenza-like illness (ILI) at sentinel hospitals and primary healthcare clinics in five provinces of South Africa since 2009. In March 2020, the NICD began testing surveillance samples for SARS-CoV-2 to understand the burden of COVID-19. To further evaluate the community burden of disease it is important to adjust the COVID-19 incidence obtained from facility-based surveillance by the healthcare utilization patterns of the sentinel site catchment population. We describe healthcare utilization patterns within the catchment areas of sentinel site healthcare facilities (Edendale Hospital and Edendale Gateway Clinic in Pietermaritzburg, KwaZulu-Natal Province, Mitchell's Plain Hospital and Mitchell's Plain Clinic in Mitchell's Plain, Western Cape Province, and Tshepong Hospital, Klerksdorp Hospital and Jouberton Clinic in Klerksdorp, North West Province).

## Materials and methods

### Study design and population

We conducted a cross-sectional HUS in households in three of the communities serviced by facilities where severe respiratory illness (SRI) and influenza-like illness (ILI) surveillance was conducted [8, 9], namely Mitchell's Plain, Pietermaritzburg, and Klerksdorp, using a one-stage cluster sampling design. Mitchell's Plain is a large township in the City of Cape Town, Western Cape Province, Pietermaritzburg is the capital and second largest city in KwaZulu-Natal Province, and Klerksdorp is the largest city in North West Province. Study sites were selected based on having long-term surveillance data available from both primary healthcare clinic (ILI surveillance) and hospital (SRI surveillance) facilities.

### Sample size

The sample size was calculated for a one-stage cluster sampling design, with a 95% confidence interval, 10% precision, a 50% expected healthcare seeking among individuals reporting a

severe respiratory illness within a predefined period of time in the community and an assumed design effect of 1.5 (household cluster). The sample size was calculated to be 144 individuals reporting a severe respiratory illness in the selected households. Based on data obtained from population-based hospital surveillance and the use of healthcare utilization surveys previously conducted in the target communities, the annual cumulative incidence of severe respiratory illness in these communities is estimated to be 2 per 100 population; therefore, we aimed to interview 7,200 individuals (i.e., 144/0.02) in each community. Assuming an average household size of 3 members, we aimed to enrol 2,400 households in each community. We accounted for a 20% household refusal rate.

## Selection and enrolment of households

Households were identified using randomly selected global positioning system (GPS) coordinates. The boundaries of each catchment area were delineated on aerial maps available from Google Earth or the local municipality. Non-residential areas such as parks, industrial areas and sports complexes were excluded. In township areas, we used aerial images of the townships to create polygons corresponding to the townships. Within each township polygon, we randomly sampled geographic coordinates where the number of coordinates was proportional to the population of the township. The household closest (within 30 meters) to each random geographic coordinate was approached for the survey. Fieldworker teams visited each selected household up to three times on separate days or times as needed. A household was excluded if the head of the household/primary caregiver was unavailable after three visits on separate days or times or declined participation in the study. Individuals of all ages that lived in recruited households since 1st March 2020 were eligible to be enrolled in the study, while individuals with unknown residence or residence outside of the catchment area were excluded. Additional GPS co-ordinates were generated at the start of the study and replacement households were visited according to the order on the list.

## Data collection

Field workers administered structured questionnaires electronically using Research Electronic Data Capture (REDCap, Vanderbilt University, USA) with the primary caregiver of the household, to gather information on household demographics and screening of household members for symptoms for SRI (sudden onset or worsening fever with cough, and difficulty breathing lasting between 2 and 30 days, or diagnosed with pneumonia) since the beginning of March 2020, or ILI (sudden onset or worsening fever with cough) in the past 30 days. SRI and ILI case definitions were adapted from previous healthcare utilization surveys [2, 4, 5]. After providing written informed consent, household members identified by the primary caregiver as having SRI and/or ILI were interviewed by the fieldworkers. If the participant reported >1 respiratory illness episode, information was collected on the most recent episode of respiratory illness. Demographic information as well as information on underlying illnesses (including tuberculosis (current or previous), asthma, diabetes, chronic heart disease, chronic lung disease, hypertension and cancer), symptoms experienced and healthcare seeking for the reported ILI/SRI episode (clinic, hospital, traditional healer, religious leader, friend/relative, community health worker or pharmacy) were collected from participants. Height and weight were measured to calculate the body mass index (BMI) for individuals aged ≥5 years. BMI could not be calculated for children aged <5 years as age in month units was not available, and was classified as unknown. Participants were asked if they had been previously tested for SARS-CoV-2 infection and the result thereof. If the household member was aged <18 years, information was obtained from the child's parent/guardian. Fieldworkers also interviewed primary

caregivers to collect healthcare seeking information for household members that had died from any cause in the period since beginning of March 2020.

## Data analysis

Body mass index (BMI) was calculated using participants measured height and weight, and categorised using WHO standards [10, 11]. Socioeconomic status (SES) was measured using a standardised set of questions described in previous research [12]. Responses to SES questions were summed, a score created and categorised into three levels (low, medium and high). Crowding in the household was defined as a mean of >2 individuals per sleeping room. Continuous variables were summarized using median and interquartile ranges (IQR). Categorical variables were summarized using frequency distributions and compared using Pearson's Chi-squared test. Hierarchical multivariable logistic regression, controlling for site and household (within site) clustering, was used to identify factors associated with healthcare utilization among individuals that reported illness, starting with all variables that were significant at p-values <0.2 on univariate analysis and dropping non-significant factors with manual stepwise backward selection. All 2-way interactions were evaluated. Two-sided p-values <0.05 were considered significant. Analysis was performed using Stata 14.1® (StataCorp LP, College Station, United States of America).

## Ethics

This study was approved by the University of the Witwatersrand (M200861) Human Research Ethics Committee; the U.S. Centers for Disease Control and Prevention Institutional Review Board (7322) relied on the local ethics committee. The study was also approved by the respective community and provincial research committees. The findings and conclusions in this report are those of the authors and do not necessarily represent the official position of the CDC.

## Results

### Description of study population

From November 2020 through April 2021, 8,690 households were visited, of which 7,032 (80.9%) had the primary caregiver available to be interviewed at the time of the visit (Fig 1). In total, 5,804/7,032 (82.5%) of approached households were enrolled in the HUS; 2,383 (41.1%) in Pietermaritzburg, 1,985 (34.2%) in Klerksdorp and 1,436 (24.7%) in Mitchell's Plain (Table 1). Enrolled households had a median of four household members (interquartile range (IQR) 3–6 individuals) and a median of five rooms (IQR 4–6 rooms). Overall, 31.1% of households were considered crowded (24.4% in Pietermaritzburg, 32.4% in Klerksdorp and 40.4% in Mitchell's Plain, P<0.001) and 31.9% to have low SES (35.3% in Pietermaritzburg, 18.5% in Klerksdorp and 44.6% in Mitchell's Plain, P<0.001). Almost all households (90.8%) had piped water to the house as their source of drinking water and electricity (95.3%) used for cooking.

Of 23,387 individuals living in the enrolled households, information on respiratory symptoms since the start of the COVID-19 epidemic in South Africa was available for 23,003 (98.4%); 8,154 (35.4%) in Pietermaritzburg, 7,371 (32.0%) in Klerksdorp and 7,478 (32.5%) in Mitchell's Plain (Table 1). Enrolled participants had a median age of 29 years (IQR 15–46 years and range 0–105 years), and 56.0% were female. Overall, 6.0% of individuals reported to be living with HIV (8.9% in Pietermaritzburg, 8.0% in Klerksdorp and 1.0% in Mitchell's Plain, P<0.001) and 14.3% to have other underlying illnesses, of which the most common were: hypertension (8.9%), diabetes (3.4%), asthma (2.2%) and tuberculosis (1.6%).

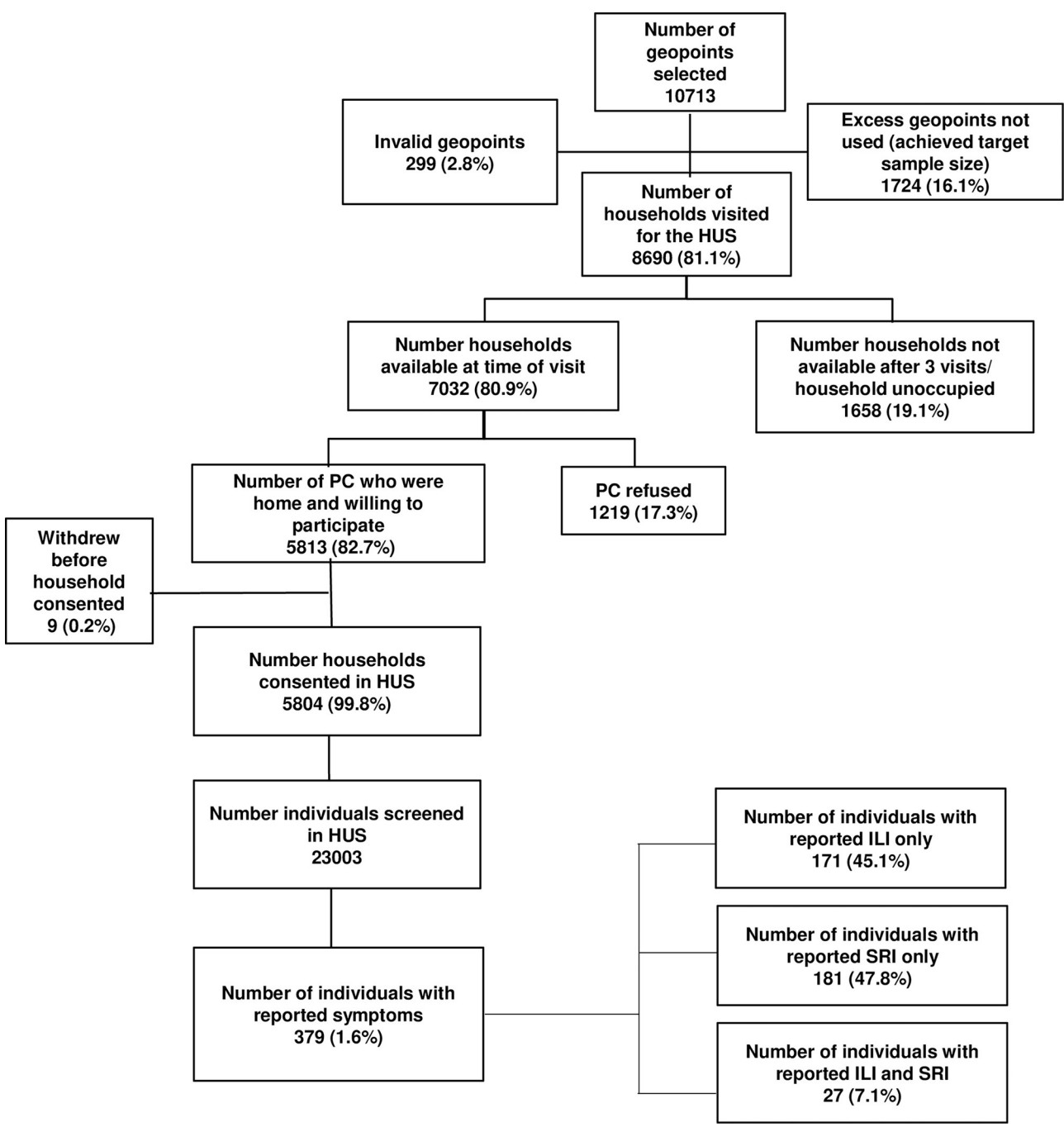

**Fig 1. Flowchart of household and participant enrolment in the healthcare utilization survey in three communities in South Africa, Healthcare Utilization and Seroprevalence (HUTS) study, November 2020 –April 2021.** PC: Primary caregiver, HUS: Healthcare utilization survey.

### Description of individuals reporting symptoms

Among 23,003 individuals interviewed in the HUS, 379 (1.6%) reported having had SRI since March 2020 and/or ILI in the past 30 days, with differences by site (168/8,154 (2.1%) in Pieter-maritzburg, 111/7,371 (1.5%) in Klerksdorp and 100/7478 (1.3%) in Mitchell's Plain, P = 0.001) (Table 1). Among individuals with respiratory symptoms, 171/379 (45.1%) reported

**Table 1. Demographic characteristics of households and participants by site, Healthcare Utilization and Seroprevalence (HUTS) study, South Africa, November 2020 –April 2021.**

| Characteristic | Overall n (%) or median (IQR) | Pietermaritzburg n (%) or median (IQR) | Klerksdorp n (%) or median (IQR) | Mitchell's Plain n (%) or median (IQR) | P-value |
|---|---|---|---|---|---|
| **Household level characteristics** | **N = 5804** | **N = 2383** | **N = 1985** | **N = 1436** | |
| Month of enrolment | N = 5804 | N = 2383 | N = 1985 | N = 1436 | <0.001 |
| November 2020 | 115 (2.0) | 78 (3.3) | 37 (1.9) | (0.0) | |
| December 2020 | 370 (6.4) | 177 (7.4) | 77 (3.9) | 116 (8.1) | |
| January 2021 | 703 (12.1) | 470 (19.7) | 105 (5.3) | 128 (8.9) | |
| February 2021 | 921 (15.9) | 335 (14.1) | 292 (14.7) | 294 (20.5) | |
| March 2021 | 2036 (35.1) | 631 (36.5) | 808 (40.7) | 597 (41.6) | |
| April 2021 | 1659 (28.6) | 692 (29.0) | 666 (33.6) | 301 (21.0) | |
| Type of house | N = 5797 | N = 2380 | N = 1985 | N = 1432 | <0.001 |
| House/Flat | 5204 (89.8) | 2162 (90.8) | 1700 (85.6) | 1342 (93.7) | |
| Traditional house | 181 (3.1) | 171 (7.2) | 7 (0.4) | 3 (0.2) | |
| Informal house/shack | 395 (6.8) | 39 (1.6) | 271 (13.7) | 85 (5.9) | |
| Other | 17 (0.3) | 8 (0.3) | 7 (0.4) | 2 (0.1) | |
| Median number of household members | 4 (3–6) | 4 (2–6) | 4 (2–5) | 6 (4–8) | |
| Number of household members | N = 5804 | N = 2383 | N = 1985 | N = 1436 | <0.001 |
| <3 | 1431 (24.7) | 608 (25.5) | 611 (30.8) | 212 (14.8) | |
| 3–5 | 2491 (42.9) | 1066 (44.7) | 932 (47.0) | 493 (34.3) | |
| 6–10 | 1627 (28.0) | 611 (25.6) | 413 (20.8) | 603 (42.0) | |
| >10 | 255 (4.4) | 98 (4.1) | 29 (1.5) | 128 (8.9) | |
| Median number of rooms | 5 (4–6) | 5 (4–7) | 4 (3–5) | 6 (5–6) | |
| Number of rooms | N = 5804 | N = 2383 | N = 1985 | N = 1436 | <0.001 |
| 1–4 | 2406 (41.5) | 877 (36.8) | 1252 (63.1) | 277 (19.3) | |
| 5–9 | 3209 (55.3) | 1383 (58.0) | 695 (35.0) | 1131 (78.8) | |
| ≥10 | 189 (3.3) | 123 (5.2) | 38 (1.9) | 28 (2.0) | |
| Crowding[a] | 1805 (31.1) | 582 (24.4) | 643 (32.4) | 580 (40.4) | <0.001 |
| Socioeconomic status | N = 5804 | N = 2383 | N = 1985 | N = 1436 | <0.001 |
| High | 1971 (34.0) | 683 (28.7) | 922 (46.5) | 366 (25.5) | |
| Medium | 1984 (34.2) | 859 (36.1) | 696 (35.1) | 429 (29.9) | |
| Low | 1849 (31.9) | 841 (35.3) | 367 (18.5) | 641 (44.6) | |
| Source of drinking water | N = 5795 | N = 2378 | N = 1983 | N = 1434 | <0.001 |
| Tap to house | 5260 (90.8) | 2183 (91.8) | 1731 (87.3) | 1346 (93.9) | |
| Communal tap | 414 (7.1) | 154 (6.5) | 203 (10.2) | 57 (4.0) | |
| Other | 121 (2.1) | 41 (1.7) | 49 (2.5) | 31 (2.2) | |
| Main fuel for cooking | N = 5611 | N = 2373 | N = 1808 | N = 1430 | <0.001 |
| Electricity | 5348 (95.3) | 2353 (99.2) | 1755 (97.1) | 1240 (86.7) | |
| Gas | 233 (4.2) | 12 (0.5) | 41 (2.3) | 180 (12.6) | |
| Wood/Paraffin/Other | 30 (0.5) | 12 (0.7) | 12 (0.7) | 10 (0.7) | |
| **Individual level characteristics** | **N = 23003** | **N = 8154** | **N = 7371** | **N = 7478** | |

*(Continued)*

**Table 1.** (Continued)

| Characteristic | Overall n (%) or median (IQR) | Pietermaritzburg n (%) or median (IQR) | Klerksdorp n (%) or median (IQR) | Mitchell's Plain n (%) or median (IQR) | P-value |
|---|---|---|---|---|---|
| Month of enrolment | N = 23003 | N = 8154 | N = 7371 | N = 7478 | <0.001 |
| November 2020 | 412 (1.8) | 279 (3.4) | 133 (1.8) | 0 (0.0) | |
| December 2020 | 1356 (5.9) | 694 (8.5) | 241 (3.3) | 421 (5.6) | |
| January 2021 | 2604 (11.3) | 1735 (21.3) | 325 (4.4) | 544 (7.3) | |
| February 2021 | 3707 (16.1) | 1229 (15.1) | 1051 (14.3) | 1427 (19.1) | |
| March 2021 | 8953 (38.9) | 2320 (28.5) | 3138 (42.6) | 3495 (46.7) | |
| April 2021 | 5971 (26.0) | 1897 (23.3) | 2483 (33.7) | 1591 (21.3) | |
| Median age (years) | 29 (15–46) | 30 (16–46) | 25 (12–43) | 32 (17–48) | |
| Age group (years) | N = 23003 | N = 8154 | N = 7371 | N = 7478 | <0.001 |
| <5 | 1331 (5.8) | 304 (3.7) | 672 (9.1) | 355 (4.8) | |
| 5–12 | 3402 (14.8) | 1142 (14.0) | 1311 (17.8) | 949 (12.7) | |
| 13–18 | 2627 (11.4) | 954 (11.7) | 914 (12.4) | 759 (10.2) | |
| 19–24 | 2345 (10.2) | 899 (11.0) | 694 (9.4) | 752 (10.1) | |
| 25–39 | 5790 (25.2) | 2171 (26.6) | 1651 (22.4) | 1968 (26.3) | |
| 40–59 | 5035 (21.9) | 1813 (22.2) | 1393 (18.9) | 1829 (24.5) | |
| ≥60 | 2473 (10.8) | 871 (10.7) | 736 (10.0) | 866 (11.6) | |
| Female sex | N = 22990 | N = 8147 | N = 7365 | N = 7478 | <0.001 |
| | 12873 (56.0) | 4748 (58.3) | 3956 (53.7) | 4169 (55.8) | |
| Reported HIV status | N = 22761 | N = 8038 | N = 7282 | N = 7441 | <0.001 |
| Living with HIV | 1368 (6.0) | 714 (8.9) | 579 (8.0) | 75 (1.0) | |
| Other underlying illness[b] | N = 22823 | N = 8066 | N = 7304 | N = 7453 | <0.001 |
| | 3254 (14.3) | 1226 (15.2) | 925 (12.7) | 1103 (14.8) | |
| Previously tested for SARS-CoV-2 | N = 22951 | N = 8143 | N = 7346 | N = 7462 | <0.001 |
| | 1393 (6.1) | 588 (7.2) | 345 (4.7) | 460 (6.2) | |
| Lab-confirmed SARS-CoV-2 infection | N = 1348 | N = 559 | N = 335 | N = 454 | 0.698 |
| | 230 (17.1) | 92 (16.5) | 55 (16.4) | 83 (18.3) | |
| Reported respiratory symptoms | N = 23003 | N = 8154 | N = 7371 | N = 7478 | 0.001 |
| | 379 (1.6) | 168 (2.1) | 111 (1.5) | 100 (1.3) | |
| Respiratory illness category | N = 379 | N = 168 | N = 111 | N = 100 | <0.001 |
| Influenza-like illness | 171 (45.1) | 74 (44.1) | 81 (73.0) | 16 (16.0) | |
| Severe respiratory illness | 181 (47.8) | 82 (48.8) | 24 (21.6) | 75 (75.0) | |
| ILI and SRI | 27 (7.1) | 12 (7.1) | 6 (5.4) | 9 (9.0) | |

[a] Crowding defined as >2 household members per sleeping room

[b] Underlying illness includes current/previous tuberculosis, asthma, diabetes, chronic heart disease, chronic lung disease, hypertension and cancer

ILI, 181/379 (47.8%) reported SRI and 27/379 (7.1%) reported both ILI and SRI. The prevalence of any respiratory illness differed by age group: 24/1,330 (1.8%) in <5 years, 18/3,401 (0.5%) in 5–12 years, 13/2,625 (0.5%) in 13–18 years, 41/2,337 (1.8%) in 19–24 years, 103/5,777 (1.8%) in 25–39 years, 113/5,012 (2.3%) in 40–59 years and 67/2,466 (2.7%) in ≥60 years (P<0.001).

The majority of individuals that reported having ILI only were adults aged >18 years (134/171, 78.4%) and females (109/171, 63.7%). Among individuals with ILI with responses recorded, the most commonly reported symptoms were cough (120/153, 78.4%), fever (70/150, 46.7%), fatigue (50/134, 37.3%), runny nose (5/153, 36.0%), sore throat (47/134, 35.1%) and headache (45/134, 33.6%).

Similarly, the majority of individuals who reported having SRI only were adults aged >18 years (166/181, 91.7%) and females (116/181, 64.1%). Among individuals with SRI, the most commonly reported symptoms were difficulty breathing (106/151, 70.2%), fever (94/142, 66.2%), cough (96/149, 64.4%) and fatigue (89/144, 61.8%).

## Healthcare seeking behaviour

For individuals where information was available, 170/316 (53.8%, 95% CI 48.1% - 59.4%) of individuals with any respiratory illness sought medical care: 62/152 (40.8%, 95% CI 32.9% - 49.6%) of individuals with ILI only and 102/143 (71.3%, 95% CI 63.2% - 78.6%) with SRI only (P<0.001). The proportion of individuals that sought medical care varied by site (97/155 (62.6%) in Pietermaritzburg, 36/96 (37.5%) in Klerksdorp and 37/65 (56.9%) in Mitchell's Plain, P<0.001). Of individuals with ILI that sought medical care, 6/61 (9.8%) were admitted to hospital, while 33/101 (32.7%) with SRI were admitted to hospital. Where information was available, the most common reason for individuals not seeking medical care for their illness (ILI and/or SRI) was that they did not think they were sick enough (91/129, 71.0%) to need medical care. Some other reasons for not seeking care were that the individual was not able to get to a facility due to lack or cost of transport (14/129, 10.9%), self-medicated (9/129, 7.0%) or did not think it was safe to visit a healthcare facility (6/129, 4.7%). Among individuals with respiratory illness, 64.7% (205/317) reported wearing a mask while they were ill, and 37.1% (118/318) self-isolated.

The proportion of individuals with ILI that sought care for their illness was higher among children aged <5 years (10/19, 52.6%) and adults aged ≥60 years (13/20, 65.0%) compared to other age groups, underweight (4/9, 44.4%) and obese (22/43, 51.2%) individuals compared to normal weight and overweight, individuals with underlying illness (29/48, 60.4%) compared to those with no underlying illness, and with symptom duration ≥10 days (22/29, 75.9%) compared to those with symptom duration <10 days (Table 2). For individuals with data available, individuals sought care predominantly at a public primary healthcare clinic (34/60, 56.7%) or private clinic/general practitioner (26/61, 42.6%), with a smaller proportion seeking care at a public (9/60, 15.0%) or private (7/61, 11.5%) hospital. A small proportion (8/62, 12.9%) of individuals reported having visited a pharmacy, with only 1.6% (1/62) visiting a religious leader and no individuals reported visiting a traditional healer. Of the 50 participants with ILI who responded that they sought care and had information on number of healthcare visits, the majority (39/50, 78.0%) had one healthcare visit with 12.0% (6/50) having two, and 10.0% (5/50) having ≥3 visits for the same episode of illness respectively. Among individuals with ILI that reported the name of the facility visited, 20.6% (7/34) visited a surveillance site facility: 12.5% (2/16) in Pietermaritzburg (Edendale Hospital), 35.7% (5/14) in Klerksdorp (n = 4 at Tshepong Hospital and n = 1 at Klerksdorp Hospital), and 0.0% (0/4) in Mitchell's Plain.

Among individuals with SRI, the proportion seeking care was higher among individuals that were underweight (7/7, 100%) or obese (43/53, 74.1%) compared to normal weight and overweight, and individuals with secondary (39/44, 88.6%) and tertiary (16/20, 80.0%) levels of education compared to lower levels of education (Table 2). Individuals sought care predominantly at a public primary healthcare clinic (49/102, 48.0%) or private clinic/general practitioner (37/102, 36.3%), with a smaller proportion seeking care at a public (27/102, 26.5%) or private (20/102, 19.6%) hospital. One quarter (27/101, 26.7%) of individuals reported having visited a pharmacy, with only 2.0% (2/102) visiting a religious leader and 2.0% (2/101) visiting a traditional healer. For participants with SRI that had information on number of healthcare visits, approximately two-thirds of individuals (51/74, 68.9%) had one healthcare visit, 20.3% (15/74) had two visits and 10.8% (8/74) had ≥3 visits for the same illness episode. Among

**Table 2. Healthcare utilization among individuals reporting respiratory illness by syndrome, Healthcare Utilization and Seroprevalence (HUTS) study, South Africa, November 2020 – April 2021 (N = 316).**

| | | Any respiratory illness (N = 316) [a] | Influenza-like illness only (N = 152) | P-value | Severe respiratory illness only (N = 143) | P-value |
|---|---|---|---|---|---|---|
| **Variable** | | n/N (%) | n/N (%) | | n/N (%) | |
| **Site** | Pietermaritzburg | 97/155 (62.6) | 34/68 (50.0) | 0.005 | 62/76 (81.6) | 0.009 |
| | Klerksdorp | 36/96 (37.5) | 20/72 (27.8) | | 13/19 (68.4) | |
| | Mitchell's Plain | 37/65 (56.9) | 8/12 (66.7) | | 27/48 (56.3) | |
| **Age group (years)** | <5 | 13/23 (56.5) | 10/19 (52.6) | 0.030 | 3/3 (100.0) | 0.077 |
| | 5–18 | 7/24 (29.2) | 3/14 (21.4) | | 3/8 (37.5) | |
| | 19–59 | 115/214 (53.7) | 36/99 (36.4) | | 75/100 (75.0) | |
| | ≥60 | 35/55 (63.6) | 13/20 (65.0) | | 21/32 (65.6) | |
| **Sex** | Male | 62/113 (54.9) | 18/53 (34.0) | 0.210 | 41/52 (78.9) | 0.133 |
| | Female | 108/203 (53.2) | 44/99 (44.4) | | 61/91 (67.0) | |
| **Highest education level** | None/some primary | 28/63 (44.4) | 14/38 (36.8) | 0.637 | 13/22 (59.1) | 0.008 |
| | Primary | 47/93 (50.5) | 20/49 (40.8) | | 25/39 (64.1) | |
| | Secondary | 57/94 (60.6) | 15/41 (36.6) | | 39/44 (88.6) | |
| | Tertiary | 27/42 (64.3) | 11/21 (52.4) | | 16/20 (80.0) | |
| | Unknown | 11/24 (45.8) | 2/3 (66.7) | | 9/18 (50.0) | |
| **Reported HIV status** | Not living with HIV | 142/272 (52.2) | 53/131 (40.5) | 0.422 | 83/121 (68.6) | 0.215 |
| | Living with HIV | 26/40 (65.0) | 9/19 (47.4) | | 17/20 (85.0) | |
| | Unknown | 2/4 (50.0) | 0/2 (0.0) | | 2/2 (100) | |
| **BMI [b]** | Underweight | 12/19 (63.2) | 4/9 (44.4) | 0.003 | 7/7 (100.0) | 0.023 |
| | Normal weight | 24/73 (32.9) | 6/35 (17.1) | | 16/29 (55.2) | |
| | Overweight | 28/60 (46.7) | 9/29 (31.0) | | 18/29 (62.1) | |
| | Obese | 67/106 (63.2) | 22/43 (51.2) | | 43/58 (74.1) | |
| | Unknown | 39/58 (67.2) | 21/36 (58.3) | | 18/20 (90.0) | |
| **Other underlying illness [c]** | No | 92/201 (45.8) | 32/102 (31.4) | 0.003 | 57/83 (68.7) | 0.409 |
| | Yes | 77/113 (68.1) | 29/48 (60.4) | | 45/60 (75.0) | |
| | Unknown | 1/2 (50.0) | 1/2 (50.0) | | 0/0 (0.0) | |
| **Duration of symptoms (days)** | <5 | 27/79 (34.2) | 16/60 (26.7) | <0.001 | 10/15 (66.7) | 0.016 |
| | 5–9 | 24/42 (57.1) | 8/18 (44.4) | | 12/16 (75.0) | |
| | ≥10 | 71/87 (81.6) | 22/29 (75.9) | | 49/58 (84.5) | |
| | Unknown | 48/108 (44.4) | 16/45 (35.6) | | 31/54 (57.4) | |
| **Socioeconomic status** | High | 28/57 (49.1) | 15/30 (50.0) | 0.275 | 12/20 (60.0) | 0.188 |
| | Medium | 51/99 (51.5) | 10/33 (30.3) | | 39/58 (67.2) | |
| | Low | 91/160 (56.9) | 37/89 (41.6) | | 51/65 (78.5) | |

[a] Reported respiratory illness (SRI only, ILI only or both SRI and ILI) for individuals where information on healthcare seeking was available.

[b] Body Mass Index (BMI) calculated for individuals aged ≥5 years

[c] Underlying illness includes current/previous tuberculosis, asthma, diabetes, chronic heart disease, chronic lung disease, hypertension and cancer

individuals with SRI that reported the name of the facility visited, 18/54 (33.3%) (18/54) attended a SRI surveillance site: 25.7% (9/35) in Pietermaritzburg (n = 3 at Edendale Gateway Clinic and n = 6 at Edendale Hospital), 50.0% (4/8) in Klerksdorp (Tshepong Hospital), and 45.5% (5/11) in Mitchell's Plain (Mitchell's Plain Clinic).

One-third (123/377) of individuals with respiratory illness reported having had a COVID-19 test: 23.4% (40/171) of individuals with ILI and 41.9% (75/179) with SRI. Among those with

a COVID-19 test result, 56.7% (68/120) tested positive: 37.8% (14/37) of individuals with ILI and 72.0% (54/75) with SRI.

## Factors associated with medically attended illness

On univariate analysis, respiratory illness was more likely to be medically attended at the Pietermaritzburg site (OR 3.8, 95% CI 1.6–9.1, compared to Klerksdorp), among children aged <5 years (OR 9.5, 95% CI 1.1–78.7) and individuals aged ≥60 years (OR 6.2, 95% CI 1.2–32.2 compared to 5–18 years), among individuals that were underweight (OR 5.1, 95% CI 1.1–22.5) or obese (OR 4.6, 95% CI 1.7–12.0, compared to normal weight), among individuals with underlying illness (OR 3.5, 95% CI 1.7–7.0), with severe illness (OR 6.1, 95% CI 2.1–17.4, compared to mild illness), and with symptom duration ≥10 days (OR 11.3, 95% CI 3.8–33.6, compared to <5 days) (Table 3).

On multivariable analysis, respiratory illness was more likely to be medically attended for individuals at the Pietermaritzburg site (aOR 3.2, 95% CI 1.1–9.5, compared to Klerksdorp), individuals that were underweight (aOR 11.5, 95% CI 1.5–90.2, compared to normal weight), that had underlying illness (aOR 3.2, 95%CI 1.2–8.5), that had severe illness (aOR 4.8, 95% CI 1.6–14.3) and those with symptom duration ≥10 days (aOR 7.9, 95% CI 2.1–30.2, compared to <5 days) (Table 3).

## Description of individuals that died

Among 5794 households for which data was available, 127 (2.2%) reported having a household member die since March 2020 (2.1% in Pietermaritzburg, 2.3% in Klerksdorp and 2.2% in Mitchell's Plain, P = 0.839). The majority of households with a death reported one member (121/127, 95.3%) having died, but 4.7% (6/127) of households with a death reported two members having died. Additional information was provided for 81/133 (61%) individuals that had died: median age was 60 years (IQR 49–69) and 53.1% (43/81) were female. Among individuals that died, 70.4% (57/81) were reported to have died from an illness: 18/57 (31.6%), 20/57 (35.1%) and 30/57 (52.6%) had symptoms of fever, cough and difficulty breathing respectively. Almost all (72/81, 88.9%) individuals that died sought medical care: 51/72 (70.8%) at a hospital, 13/72 (18.1%) at a clinic, 6/72 (8.3%) at a private doctor, 1/72 (1.4%) at a traditional healer and 1/72 (1.4%) sought other care. Fifty individuals (61.7%) died in hospital, 30/81 (37.0%) at home and 1/81 (1.2%) on the way to hospital. For 39 individuals for whom the hospital was known, 27/39 (69.2%) died at a sentinel site hospital.

## Discussion

In a household survey conducted in three communities in South Africa after the first two waves of the COVID-19 pandemic (dominated by the ancestral and Beta variants of concern, respectively), 1.6% of individuals reported experiencing respiratory illness. ILI was reported by 0.7% of individuals during the 30 days prior to the interview and severe respiratory illness by 0.8% of individuals since March 2020. The majority of individuals reporting ILI (78.4%) and SRI (91.7%) were older than 18 years. Healthcare utilization was higher for individuals with SRI (71.3%) than those with ILI (40.8%). Respiratory illness was more likely to be medically attended in individuals at the Pietermaritzburg site, underweight individuals, individuals that had underlying illness or severe illness and those with symptom duration ≥10 days. Among interviewed households, 2.2% reported having had at least one household member die since the start of the COVID-19 pandemic.

In our study in the first year of the COVID-19 pandemic, 0.7% of individuals reported having experienced ILI in the past 30 days and 0.8% experienced SRI since March 2020. During this time, South Africa experienced two epidemic waves dominated by the ancestral virus and

**Table 3. Factors associated with healthcare utilization among individuals reporting respiratory illness (ILI and/or SRI), Healthcare Utilization and Seroprevalence (HUTS) study, South Africa, November 2020 –April 2021.**

| Variable | | Healthcare Utilization | Univariate[a] | | Multivariable[a,b] | |
|---|---|---|---|---|---|---|
| | | n/N (%) | OR (95% CI) | P-value | aOR (95% CI) | P-value |
| **Site** | Pietermaritzburg | 97/155 (62.6) | 3.8 (1.6–9.1) | **0.002** | 3.2 (1.1–9.5) | **0.040** |
| | Klerksdorp | 36/96 (37.5) | Ref | - | Ref | - |
| | Mitchell's Plain | 37/65 (56.9) | 2.5 (0.9–6.6) | 0.067 | 1.3 (0.4–4.6) | 0.668 |
| **Month of enrolment** | November 2020 | 11/19 (57.9) | Ref | - | | |
| | December 2020 | 16/29 (55.2) | 0.6 (0.1–3.9) | 0.629 | | |
| | January 2021 | 28/57 (49.1) | 0.5 (0.1–2.4) | 0.349 | | |
| | February 2021 | 37/60 (61.7) | 1.2 (0.2–6.1) | 0.851 | | |
| | March 2021 | 62/108 (57.4) | 1.0 (0.2–5.0) | 0.956 | | |
| | April 2021 | 16/43 (37.2) | 0.4 (0.1–2.4) | 0.317 | | |
| **Age group (years)** | <5 | 13/23 (56.5) | 9.5 (1.1–78.7) | **0.037** | 6.4 (0.4–97.9) | 0.185 |
| | 5–18 | 7/24 (29.2) | Ref | - | Ref | - |
| | 19–59 | 115/214 (53.7) | 3.7 (0.9–15.9) | 0.077 | 2.3 (0.5–11.9) | 0.314 |
| | ≥60 | 35/55 (63.6) | 6.2 (1.2–32.2) | **0.030** | 2.2 (0.4–13.7) | 0.381 |
| **Sex** | Male | 62/113 (54.9) | Ref | - | | |
| | Female | 108/203 (53.2) | 0.8 (0.4–1.6) | 0.548 | | |
| **Highest education level** | None/some primary | 28/63 (44.4) | Ref | - | | |
| | Primary | 47/93 (50.5) | 1.0 (0.4–2.6) | 0.959 | | |
| | Secondary | 57/94 (60.6) | 1.6 (0.6–4.0) | 0.347 | | |
| | Tertiary | 27/42 (64.3) | 1.5 (0.5–5.2) | 0.496 | | |
| | Unknown | 11/24 (45.8) | 0.5 (0.1–2.5) | 0.419 | | |
| **Reported HIV status** | Not living with HIV | 142/272 (52.2) | Ref | - | | |
| | Living with HIV | 26/40 (65.0) | 2.0 (0.7–5.6) | 0.209 | | |
| | Unknown | 2/4 (50.0) | 0.2 (0.0–5.5) | 0.375 | | |
| **BMI[c]** | Underweight | 12/19 (63.2) | 5.1 (1.1–22.5) | **0.033** | 11.5 (1.5–90.2) | **0.020** |
| | Normal weight | 24/73 (32.9) | Ref | - | Ref | - |
| | Overweight | 28/60 (46.7) | 2.1 (0.8–5.7) | 0.148 | 1.5 (0.4–4.7) | 0.536 |
| | Obese | 67/106 (63.2) | 4.6 (1.7–12.0) | **0.002** | 3.0 (1.0–9.4) | 0.061 |
| | Unknown | 39/58 (67.2) | 7.6 (2.3–25.6) | **0.001** | 7.5 (1.4–38.8) | **0.017** |
| **Other underlying illness[d]** | No | 92/201 (45.8) | Ref | - | Ref | - |
| | Yes | 77/113 (68.1) | 3.5 (1.7–7.0) | **<0.001** | 3.2 (1.2–8.5) | **0.018** |
| | Unknown | 1/2 (50.0) | 0.9 (0.0–26.3) | 0.957 | 1.2 (0.0–41.1) | 0.917 |
| **Severity of illness** | ILI (mild) | 62/152 (40.8) | Ref | - | Ref | - |
| | SRI (severe) | 102/143 (71.3) | 6.1 (2.1–17.4) | **0.001** | 4.8 (1.6–14.3) | **0.005** |
| **Duration of symptoms (days)** | <5 | 27/79 (34.2) | Ref | - | Ref | - |
| | 5–9 | 24/42 (57.1) | 2.7 (0.9–7.5) | 0.063 | 1.3 (0.3–4.8) | 0.733 |
| | ≥10 | 71/87 (81.6) | 11.3 (3.8–33.6) | **<0.001** | 7.9 (2.1–30.2) | **0.003** |
| | Unknown | 48/108 (44.4) | 1.4 (0.6–3.2) | 0.374 | 1.1 (0.4–3.3) | 0.836 |
| **Socioeconomic status** | High | 28/57 (49.1) | Ref | - | | |
| | Medium | 51/99 (51.5) | 0.9 (0.3–2.6) | 0.904 | | |
| | Low | | | | | |

[a] Adjusted for clustering by site and household

[b] Variables assessed in multivariable model: site, age group, BMI, other underlying illness, severity of illness, duration of symptoms

[c] Body Mass Index (BMI) calculated for individuals aged ≥5 years

[d] Underlying illness includes current/previous tuberculosis, asthma, diabetes, chronic heart disease, chronic lung disease, hypertension and cancer

Beta variants, respectively. A national lockdown was implemented from 27 March– 1 May 2020, followed by gradual and phased changing of restrictions (including non-pharmaceutical interventions such as social distancing and compulsory mask wearing) in response to the national number of cases [1]. Previous healthcare utilization surveys have been conducted in sentinel site catchment populations in order to estimate the proportion of illness not captured by routine facility-based surveillance. In a survey conducted in Pietermaritzburg in 2013, 5% of individuals reported ILI in the past 30 days and 0.5% reported pneumonia [5]. In a survey conducted in Klerksdorp in 2012, 2% of individuals reported ILI in the past 30 days and 2% reported pneumonia [4]. In our study, the proportion of individuals with ILI was lower than in previous studies. This may reflect the observed lower activity of non-SARS-CoV-2 respiratory pathogens due to the enforcement of non-pharmaceutical interventions during the pandemic [13].

During the first year of the COVID-19 pandemic, 59.2% of individuals with ILI and 28.7% with SRI reported did not seek medical care for their illness. In a previous healthcare utilization survey in Pietermaritzburg, 13% of individuals with ILI did not seek medical care [5]. In Klerksdorp in 2012, 0% and 28% of individuals with ILI and pneumonia did not seek medical care [4]. Compared to previous studies in the same areas, our study showed reduced healthcare utilization during the initial pandemic period. This reflects trends reported in many countries in which there was decrease in healthcare utilization during this period. A systematic review including data from 20 countries reported a 37% reduction in in use of healthcare services [14]. A reduction in care seeking was observed in the 18 months following the start of the pandemic in the private healthcare sector in South Africa [15]. Similar to our study, the most common reason for not seeking medical care in previous healthcare utilization surveys conducted in South Africa was individuals not feeling sick enough, and the most common facility where care was sought was public clinics [4–6]. Some individuals also reported not seeking medical care because they were not able to get transport and not feeling that it was safe.

Individuals were more likely to seek medical care if they were underweight, had underlying illness, experienced severe illness or had longer duration of symptoms. On univariate analysis, young (<5 years) and older (≥60 years) aged individuals were more likely to seek medical care, although age was not significant in the multivariable model, possibly due to small numbers. In a systematic review of the impact of COVID-19 on healthcare utilization, greater reductions in seeking medical care were observed for milder cases of illness [14]. A previous survey in Soweto in 2012, South Africa reported care seeking to have been associated with female sex and age <18 years [4].

Our study had a number of limitations. First, all respiratory illnesses were based on self-reports and a time period 30 days for ILI and approximately one year for SRI which may have been influenced by recall bias. Second, the case definitions used were syndromic and not laboratory-confirmed. These may have resulted in misclassification of ILI and SRI cases. Third, only 67% of households visited were enrolled in the study, and the results may; therefore, not be fully representative of the communities in which the surveys were conducted. Fourth, the number of individuals that did not seek healthcare for their illness is likely underestimated as individuals that had died during the study period were not included. Finally, the number of individuals reporting respiratory symptoms and seeking healthcare during the first two waves of the pandemic were small and we may have been underpowered to detect factors associated with medically attended illness such as age.

## Conclusions

Healthcare utilization surveys improve the interpretation of facility-based surveillance data by quantifying the proportion of individuals that did not seek medical care for their illness. In our

study, during the first year of the COVID-19 pandemic, only 40.8% and 71.3% of individuals with mild and severe disease respectively, sought medical care for their illness. Healthcare utilization for respiratory illness was severely lower than in surveys done in earlier years. This should be taken into account when quantifying the burden of COVID-19 disease using facility-based data.

## Acknowledgments

Contributors to the HUTS study are thanked for their inputs. These include: participants who agreed to be part of the HUTS study; the Department of Health and district counsellors for supporting this research and the Epicentre fieldwork teams.

## Author Contributions

**Conceptualization:** Nicole Wolter, Stefano Tempia, Sibongile Walaza, Jocelyn Moyes, Cheryl Cohen.

**Data curation:** Jackie Kleynhans, Sue Aitken, Jessica Yun, Tamika Fellows, Tetelo Makamadi, Renay Weiner, Cherie Cawood.

**Formal analysis:** Nicole Wolter, Stefano Tempia, Jackie Kleynhans.

**Funding acquisition:** Cheryl Cohen.

**Investigation:** Nicole Wolter, Neil Martinson, Cheryl Cohen.

**Methodology:** Nicole Wolter, Stefano Tempia, Anne von Gottberg, Sibongile Walaza, Jocelyn Moyes, Sue Aitken, Jessica Yun, Tamika Fellows, Renay Weiner, Cherie Cawood, Cheryl Cohen.

**Project administration:** Sue Aitken, Sarah Magni, Cherie Cawood, Neil Martinson, Limakatso Lebina.

**Supervision:** Nicole Wolter, Anne von Gottberg, Jinal N. Bhiman, Sue Aitken, Sarah Magni, Cherie Cawood, Neil Martinson, Cheryl Cohen.

**Writing – original draft:** Nicole Wolter.

**Writing – review & editing:** Nicole Wolter, Stefano Tempia, Anne von Gottberg, Jinal N. Bhiman, Sibongile Walaza, Jackie Kleynhans, Jocelyn Moyes, Sue Aitken, Sarah Magni, Jessica Yun, Tamika Fellows, Tetelo Makamadi, Renay Weiner, Cherie Cawood, Neil Martinson, Limakatso Lebina, Cheryl Cohen.

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
