## [Decision Letter · Decision Letter 0]

27 Jun 2023

PONE-D-23-03137Healthcare utilization during the first two waves of the COVID-19 epidemic in South Africa: a cross-sectional household surveyPLOS ONE

Dear Dr. Nicole,

Thank you for submitting your manuscript to PLOS ONE. After careful consideration, we feel that it has merit but does not fully meet PLOS ONE’s publication criteria as it currently stands. Therefore, we invite you to submit a revised version of the manuscript that addresses the points raised during the review process.

We look forward to receiving your revised manuscript.

Kind regards,

Surangi Jayakody, MBBS, MSc, MD

Academic Editor

PLOS ONE

Journal Requirements:

2. Thank you for stating the following in the Competing Interests/Financial Disclosure * (delete as necessary) section:

"We acknowledge funding from the South African Medical Research Council (https://www.samrc.ac.za/) (Reference number SHIPNCD 76756) [C. Cohen], The Wellcome Trust and the United Kingdom Foreign, Commonwealth and Development Office (https://wellcome.org/) (Grant no 221003/Z/20/Z) [C. Cohen] and United States Centers for Disease Control and Prevention (https://www.cdc.gov/) (Grant number 5 U01IP001048-05-00) [C. Cohen]. The funders had no role in study design, data collection and analysis, decision to publish, or preparation of the manuscript."

We note that you received funding from a commercial source: "Wellcome Trust"

"I have read the journal's policy and the authors of this manuscript have the following competing interests: CC has received grant support from Sanofi Pasteur, South African Medical Research Council, The Wellcome Trust and the United Kingdom Foreign, Commonwealth and Development Office, PATH, US Centers for Disease Control and Prevention (CDC) and the Bill and Melinda Gates Foundation. NW and AvG have received grant support from Sanofi Pasteur, US Centers for Disease Control and Prevention (CDC) and the Bill and Melinda Gates Foundation."

6. We note that Supplementary figures 1 and 2 in your submission contain [map, satellite] images which may be copyrighted. All PLOS content is published under the Creative Commons Attribution License (CC BY 4.0), which means that the manuscript, images, and Supporting Information files will be freely available online, and any third party is permitted to access, download, copy, distribute, and use these materials in any way, even commercially, with proper attribution. For these reasons, we cannot publish previously copyrighted maps or satellite images created using proprietary data, such as Google software (Google Maps, Street View, and Earth). For more information, see our copyright guidelines: http://journals.plos.org/plosone/s/licenses-and-copyright.

    1. You may seek permission from the original copyright holder of Supplementary figures 1 and 2 to publish the content specifically under the CC BY 4.0 license. 

Reviewers' comments:

Reviewer's Responses to Questions

**Comments to the Author**

1. Is the manuscript technically sound, and do the data support the conclusions?

Reviewer #1: Yes

Reviewer #2: Yes

2. Has the statistical analysis been performed appropriately and rigorously? 

Reviewer #1: I Don't Know

Reviewer #2: Yes

3. Have the authors made all data underlying the findings in their manuscript fully available?

Reviewer #1: No

Reviewer #2: Yes

4. Is the manuscript presented in an intelligible fashion and written in standard English?

Reviewer #1: Yes

Reviewer #2: Yes

5. Review Comments to the Author

Reviewer #1: The research team has conducted a well-planned study of good quality to understand the healthcare utilization during covid-19 pandemic. The article is well written. I suggest reviewers to consider the following questions for further clarity.

1. Please indicate how the case definition for ILI and SRI were determined with references and include this information in the Methods section.

2. The authors have mentioned BMI was calculated only for 5 years and above. Why was weight for age not considered for < 5-year-olds?

3. According to table 3, BMI was included as a dependent variable in the logistic regression analysis. How were unavailable BMI data from 1331 children <5 years compensated?

4. According to the analysis, 11.1% of those who died have not sought medical care and 37% have died at home. This could be a potential bias, underestimating the prevalence of people who did not seek healthcare during the specified time.

5. Authors conclusion that community education is essential to promote timely presentation cannot be agreed with for two main reasons. Firstly, your study did not assess the effectiveness of community education on the timely presentation for healthcare, hence is not based on your findings. Secondly, the majority of your study sample did not utilize healthcare because they thought they were not sick enough. Do authors justify seeking early care for all respiratory symptoms or how does one draw the line when to seek care? How would this align with efficient healthcare resource utilization and burden on the health system? Therefore, revising the conclusion is highly recommended.

Reviewer #2: This is an interesting and important paper which fills an information gap - the proportion of people with ILI and SRI that actually end up seeking healthcare during the COVID pandemic. It is useful evidence for interpreting sentinel site data. There are some minor clarifications to be made to further strengthen the paper.

Data available without restriction, but can only be given for researchers who meet criteria - is the first answer correct?

Abstract

Should it be only ILI (0.7%), only SRI (0.8%)? (p3, line 45)

Introduction

Is it possible to mention where the sentinel surveillance takes place (I assume “facilities” as mentioned in line 82?), to add more clarity to this paragraph? (p5, line 80)

Regarding the three sites surveyed (Mitchell’s Plain (Western Cape Province), Pietermaritzburg (KwaZulu-Natal Province), and Klerksdorp (North West Province)):

Can you provide some pertinent information about the three sites? e.g. are they similar to each other? Urban/rural? Area SES status, population density etc, so readers not familiar with the South African setting can get a better sense of each of your sites? You do explore the respondents’ characteristics by site in detail in the results, so a more general overview may be useful in the introduction. (p6, line 83)

Methods

Can you briefly describe why you choose those three communities (I understood that there are five provinces for sentinel surveillance?) (p6, line 95)

Do you have a source for these definitions of SRI and ILI? (p7/p8)

You could add a comment that it is possible to have SRI without ILI if a person reported only pneumonia (is that the case?), so that the tables are easier to understand (p8, line 132).

Healthcare seeking - what was the recall period? or was it specific to the seeking healthcare for the ILI/SRI illness? (p8, line 136)

participants’ (p8, line 146)

SES (p8, line 148) - could you include more detail on this in the supplement? Did you use PCA or some other method to create a score? Is the way you calculated SES validated in some way?

Results

Is it “enrolled [participants] had a median age.. “ (p10, line 180)

Healthcare seeking varied by site [please clarify for what]. (p11, line 206/7)

Died deaths —> died (p13, line 269)

Discussion

Could you please provide some context on the 2012/3 surveys - were they done because of some respiratory illness that occurred during that time, hence higher numbers, or was it a based on routine surveillance? (p14, line 297-300)

6. PLOS authors have the option to publish the peer review history of their article (what does this mean?). If published, this will include your full peer review and any attached files.

Reviewer #1: No

Reviewer #2: No

---

## [Author Response · Author response to Decision Letter 0]

17 Jul 2023

Thank you for the feedback from the reviewers, which has been helpful in improving the manuscript. All references to page numbers are based on the track changes version of the revised manuscript.

Journal Requirements:

Reply: The revised manuscript has been edited to meet PLOS ONE's style requirements.

2. Thank you for stating the following in the Competing Interests/Financial Disclosure * (delete as necessary) section:

"We acknowledge funding from the South African Medical Research Council (https://www.samrc.ac.za/) (Reference number SHIPNCD 76756) [C. Cohen], The Wellcome Trust and the United Kingdom Foreign, Commonwealth and Development Office (https://wellcome.org/) (Grant no 221003/Z/20/Z) [C. Cohen] and United States Centers for Disease Control and Prevention (https://www.cdc.gov/) (Grant number 5 U01IP001048-05-00) [C. Cohen]. The funders had no role in study design, data collection and analysis, decision to publish, or preparation of the manuscript."

We note that you received funding from a commercial source: "Wellcome Trust"

Reply: Cheryl Cohen received funding from the Wellcome Trust. This has already been included in the Competing Interests Statement as highlighted in yellow: ”CC has received grant support from Sanofi Pasteur, South African Medical Research Council, The Wellcome Trust and the United Kingdom Foreign, Commonwealth and Development Office, PATH, US Centers for Disease Control and Prevention (CDC) and the Bill and Melinda Gates Foundation. NW and AvG have received grant support from Sanofi Pasteur, US Centers for Disease Control and Prevention (CDC) and the Bill and Melinda Gates Foundation.”

Reply: This sentence has been added to the Competing Interests Statement as follows: CC has received grant support from Sanofi Pasteur, South African Medical Research Council, The Wellcome Trust and the United Kingdom Foreign, Commonwealth and Development Office, PATH, US Centers for Disease Control and Prevention (CDC) and the Bill and Melinda Gates Foundation. NW and AvG have received grant support from Sanofi Pasteur, US Centers for Disease Control and Prevention (CDC) and the Bill and Melinda Gates Foundation. This does not alter our adherence to PLOS ONE policies on sharing data and materials.

Reply: The amended Competing Interests Statement has been included in the cover letter, thank you.

"I have read the journal's policy and the authors of this manuscript have the following competing interests: CC has received grant support from Sanofi Pasteur, South African Medical Research Council, The Wellcome Trust and the United Kingdom Foreign, Commonwealth and Development Office, PATH, US Centers for Disease Control and Prevention (CDC) and the Bill and Melinda Gates Foundation. NW and AvG have received grant support from Sanofi Pasteur, US Centers for Disease Control and Prevention (CDC) and the Bill and Melinda Gates Foundation."

Reply: This sentence has been added to the Competing Interests Statement as follows: CC has received grant support from Sanofi Pasteur, South African Medical Research Council, The Wellcome Trust and the United Kingdom Foreign, Commonwealth and Development Office, PATH, US Centers for Disease Control and Prevention (CDC) and the Bill and Melinda Gates Foundation. NW and AvG have received grant support from Sanofi Pasteur, US Centers for Disease Control and Prevention (CDC) and the Bill and Melinda Gates Foundation. This does not alter our adherence to PLOS ONE policies on sharing data and materials.

Reply: The dataset used and analysed in this study has been uploaded to GitHub, and can be accessed using the following link: https://github.com/crdm-nicd/huts_hus_2020_2021.git. The data availability statement has also been updated.

Reply: Thank you, we have made the data available, and have updated the data availability statement as follows: “The dataset used and analysed during the current study is available at https://github.com/crdm-nicd/huts_hus_2020_2021.git”

6. We note that Supplementary figures 1 and 2 in your submission contain [map, satellite] images which may be copyrighted. All PLOS content is published under the Creative Commons Attribution License (CC BY 4.0), which means that the manuscript, images, and Supporting Information files will be freely available online, and any third party is permitted to access, download, copy, distribute, and use these materials in any way, even commercially, with proper attribution. For these reasons, we cannot publish previously copyrighted maps or satellite images created using proprietary data, such as Google software (Google Maps, Street View, and Earth). For more information, see our copyright guidelines: http://journals.plos.org/plosone/s/licenses-and-copyright.

 1. You may seek permission from the original copyright holder of Supplementary figures 1 and 2 to publish the content specifically under the CC BY 4.0 license. 

Reply: Supplementary figures 1 and 2 have been removed from the manuscript.

Reply: The reference list has been reviewed and is complete and correct.

Reviewers' comments:

5. Review Comments to the Author

Reviewer #1: The research team has conducted a well-planned study of good quality to understand the healthcare utilization during covid-19 pandemic. The article is well written. I suggest reviewers to consider the following questions for further clarity.

1. Please indicate how the case definition for ILI and SRI were determined with references and include this information in the Methods section.

Reply: The references on which the case definitions for SRI and ILI were defined have been added to the methods section, page 8 as follows: “SRI and ILI case definitions were adapted from previous healthcare utilization surveys [2,5,10].”

2. The authors have mentioned BMI was calculated only for 5 years and above. Why was weight for age not considered for < 5-year-olds?

Reply: We were not able to calculate weight for age for children aged <5 years, as this calculation requires age in months, which was not available from the dataset. Only data in years was available and therefore BMI could only be calculated for individuals aged ≥5 years. This has been added to the methods, page 8: “BMI could not be calculated for children aged <5 years as age in month units was not available, and were classified as unknown.”

3. According to table 3, BMI was included as a dependent variable in the logistic regression analysis. How were unavailable BMI data from 1331 children <5 years compensated?

Reply: Individuals for whom BMI data was unavailable (including children aged <5 years) were classified as “unknown” in tables 2 and 3 (multivariable logistic regression analysis). This has been added to the methods, page 8: “BMI could not be calculated for children aged <5 years as age in month units was not available, and were classified as unknown.”

4. According to the analysis, 11.1% of those who died have not sought medical care and 37% have died at home. This could be a potential bias, underestimating the prevalence of people who did not seek healthcare during the specified time.

Reply: We agree with the reviewer and have added the following to the limitations paragraph, page 25: “Fourth, the number of individuals that did not seek healthcare for their illness is likely underestimated as individuals that had died during the study period were not included.”

5. Authors conclusion that community education is essential to promote timely presentation cannot be agreed with for two main reasons. Firstly, your study did not assess the effectiveness of community education on the timely presentation for healthcare, hence is not based on your findings. Secondly, the majority of your study sample did not utilize healthcare because they thought they were not sick enough. Do authors justify seeking early care for all respiratory symptoms or how does one draw the line when to seek care? How would this align with efficient healthcare resource utilization and burden on the health system? Therefore, revising the conclusion is highly recommended.

Reply: As suggested by the reviewer, we have removed this sentence from the conclusions paragraph on page 25. 

Reviewer #2: This is an interesting and important paper which fills an information gap - the proportion of people with ILI and SRI that actually end up seeking healthcare during the COVID pandemic. It is useful evidence for interpreting sentinel site data. There are some minor clarifications to be made to further strengthen the paper.

Data available without restriction, but can only be given for researchers who meet criteria - is the first answer correct?

Reply: We have made the data publically available, and have updated the data availability statement as follows: “The dataset used and analysed during the current study is available at https://github.com/crdm-nicd/huts_hus_2020_2021.git”

Abstract

Should it be only ILI (0.7%), only SRI (0.8%)? (p3, line 45)

Reply: This sentence has been updated as suggested by the reviewer, abstract page 3.

Introduction

Is it possible to mention where the sentinel surveillance takes place (I assume “facilities” as mentioned in line 82?), to add more clarity to this paragraph? (p5, line 80)

Reply: This sentence has been updated as follows, page 5: “The National Institute for Communicable Diseases (NICD) has conducted syndromic surveillance for severe respiratory illness (SRI) and influenza-like illness (ILI) at sentinel hospitals and primary healthcare clinics in five provinces of South Africa since 2009”

Regarding the three sites surveyed (Mitchell’s Plain (Western Cape Province), Pietermaritzburg (KwaZulu-Natal Province), and Klerksdorp (North West Province)):

Can you provide some pertinent information about the three sites? e.g. are they similar to each other? Urban/rural? Area SES status, population density etc, so readers not familiar with the South African setting can get a better sense of each of your sites? You do explore the respondents’ characteristics by site in detail in the results, so a more general overview may be useful in the introduction. (p6, line 83)

Reply: We have added information on the three study sites in the methods section, page 6 as follows: “Mitchell’s Plain is a large township in the City of Cape Town, Western Cape Province, Pietermaritzburg is the capital and second largest city in KwaZulu-Natal Province, and Klerksdorp is the largest city in North West Province.”

Methods

Can you briefly describe why you choose those three communities (I understood that there are five provinces for sentinel surveillance?) (p6, line 95)

Reply: Sites were chosen based on having had a number of years of facility-based surveillance data available as well as having both primary healthcare clinics (ILI surveillance) and hospitals (SRI surveillance) sentinel sites. We have added the following to methods, page 6: “Study sites were selected based on having long-term surveillance data available from both primary healthcare clinic (ILI surveillance) and hospital (SRI surveillance) facilities.”

Do you have a source for these definitions of SRI and ILI? (p7/p8)

Reply: The references on which the case definitions for SRI and ILI were defined have been added to the methods section, page 8 as follows: “SRI and ILI case definitions were adapted from previous healthcare utilization surveys [2,5,10].”

You could add a comment that it is possible to have SRI without ILI if a person reported only pneumonia (is that the case?), so that the tables are easier to understand (p8, line 132).

Reply: The following has been added to methods, page 8: “If the participant reported >1 respiratory illness episode, information was collected on the most recent episode of respiratory illness.”

Healthcare seeking - what was the recall period? or was it specific to the seeking healthcare for the ILI/SRI illness? (p8, line 136)

Reply: Healthcare seeking was specific to the ILI/SRI illness episode reported by the participant. This has been clarified in methods, page 8: “Demographic information as well as information on underlying illnesses (including tuberculosis (current or previous), asthma, diabetes, chronic heart disease, chronic lung disease, hypertension and cancer), symptoms experienced and healthcare seeking for the reported ILI/SRI episode (clinic, hospital, traditional healer, religious leader, friend/relative, community health worker or pharmacy) were collected from participants.”

participants’ (p8, line 146)

Reply: This has not been changed as it is written as a plural, not as a possessive term in this context.

SES (p8, line 148) - could you include more detail on this in the supplement? Did you use PCA or some other method to create a score? Is the way you calculated SES validated in some way?

Reply: The reference used to determine SES has been added in methods, page 9: “Socioeconomic status (SES) was measured using a standardised set of questions described in previous research [13]”

Results

Is it “enrolled [participants] had a median age.. “ (p10, line 180)

Reply: Thank you for bringing this to our attention, we have added the word “participants”, page 14.

Healthcare seeking varied by site [please clarify for what]. (p11, line 206/7)

Reply: This sentence has been updated for clarification as follows, page 15: “The proportion of individuals that sought medical care varied by site (97/155 (62.6%) in Pietermaritzburg, 36/96 (37.5%) in Klerksdorp and 37/65 (56.9%) in Mitchell’s Plain, P<0.001.”

Died deaths —> died (p13, line 269)

Reply: Thank you, this has been corrected as suggested (page 22).

Discussion

Could you please provide some context on the 2012/3 surveys - were they done because of some respiratory illness that occurred during that time, hence higher numbers, or was it a based on routine surveillance? (p14, line 297-300)

Reply: The following sentence has been added to the discussion, page 23: “Previous healthcare utilization surveys have been conducted in sentinel site catchment populations in order to estimate the proportion of illness not captured by routine facility-based surveillance"

---

## [Editor Report · Decision Letter 1]

16 Aug 2023

Healthcare utilization during the first two waves of the COVID-19 epidemic in South Africa: A cross-sectional household survey

PONE-D-23-03137R1

Dear Dr.Nicole,

We’re pleased to inform you that your revised manuscript has been judged scientifically suitable for publication and will be formally accepted for publication once it meets all outstanding technical requirements.

Kind regards,

Surangi Jayakody, MBBS, MSc, MD

Academic Editor

PLOS ONE
---

## [Editor Report · Acceptance letter]

18 Aug 2023

PONE-D-23-03137R1 

Healthcare utilization during the first two waves of the COVID-19 epidemic in South Africa:
A cross-sectional household survey 

Dear Dr. Wolter:

I'm pleased to inform you that your manuscript has been deemed suitable for publication in PLOS ONE. Congratulations! Your manuscript is now with our production department. 

Kind regards, 

on behalf of

Dr Surangi Jayakody 

Academic Editor

PLOS ONE